# Discerning Transcriptomic and Biochemical Responses of *Arabidopsis thaliana* Treated with the Biofertilizer Strain *Priestia megaterium* YC4-R4: Boosting Plant Central and Secondary Metabolism

**DOI:** 10.3390/plants11223039

**Published:** 2022-11-10

**Authors:** Ana Sofia Rodrigues-dos Santos, Inês Rebelo-Romão, Huiming Zhang, Juan Ignacio Vílchez

**Affiliations:** 1Instituto de Tecnologia Química e Biológica (ITQB)-NOVA Lisboa, 2780-157 Oeiras, Portugal; 2Shanghai Center for Plant Stress Biology, CAS Center for Excellence in Molecular Plant Science, Chinese Academy of Sciences, Shanghai 200032, China; 3State Key Laboratory of Crop Stress Adaptation and Improvement, School of Life Sciences, Henan University, Kaifeng 475001, China

**Keywords:** biofertilizers, plant metabolism induction, *Priestia megaterium*, cell wall biogenesis, biochemical traits

## Abstract

As a response to the current challenges in agriculture, the application of alternatives to a more sustainable management is required. Thus, biofertilizers begin to emerge as a reliable alternative to improve crop development and resistance to stresses. Among other effects on the plant, the use of beneficial strains may cause changes in their metabolic regulation, as in cell wall biogenesis and in nutrient/ion transportation, improving their growth process. Previous works showed that inoculation with the strain *Priestia megaterium* YC4-R4 effectively promoted vegetative growth of *Arabidopsis thaliana* Col-0 plants. Hence, the present work recorded a strain-mediated induction of several pathways of the central and secondary metabolism of the plant, as the induction of lipid, cellulose, phenol, and flavonoid biosynthesis, by using transcriptomic and biochemical analyses.

## 1. Introduction

Despite the advances in agriculture management during the last few decades, we find ourselves in a scenario where productivity is beginning to be compromised. A higher input of chemical fertilizers is not generating the expected increases in production [1,2]. Apart from the biological limits of plants, soil depletion and climate change conditions are placing productivity in check throughout the world’s arable land [3]. In this context, soil health is closely linked to the biodiversity of its microbiota, and this is being compromised by the impoverishment and desertification of the soil due in part to current management. In addition, the increase in temperatures and in drought intensity are causing desertification to advance. In this sense, better and more sustainable practices in terms of land use and agricultural production are necessary to face production problems in the future [4,5,6,7]. 

Here, the use of beneficial bacteria has been shown to be a sustainable alternative to enhance crop management and soil health. They have been used to fertilize and improve plant growth (biofertilizers), to protect plants against pests and pathogens (biocontrol), or to improve their response to different environmental stresses [8,9,10]. In recent decades, more and more studies have pointed out that the beneficial microbiota has the necessary potential to make notable and positive changes in agriculture management. In the case of key crop species, improvements in growth have been reported in all key plant species for current food (i.e., corn, soybeans, wheat, beans, rice, potato, tomato, or pepper), as well as plants with growing interest, such as quinoa, chia, or oca [11,12,13,14,15]. This broad spectrum of crops shows that the use of this biotechnology may become a reliable eco-friendly and safe substitute for widespread use [16,17]. This does not implicate stopping the use of chemical fertilizers or pest controllers, but reducing them drastically and making more efficient use of them.

The main problem applying this alternative is that the effectiveness in the field is usually reduced or limited, with respect to that registered under lab- or greenhouse-controlled conditions. In this sense, the mechanisms behind the beneficial interaction must be precisely uncovered in order to apply this technology. We know that beneficial bacteria are capable of increasing nutrient accessibility to plants (P-solubilization, siderophores, and N_2_ fixation), regulating their development by controlling or producing phytohormones (ethylene, auxins, and cytokinins), boosting their defense system (abscisic acid, jasmonic acid, and salicylic acid), improving their response, and protecting against abiotic stresses (xeroprotectors, halotolerants, and osmoregulators) [18,19,20]. However, under this interaction, the plant metabolic activity can also be modulated by the bacteria. This may occur as a result of an adjustment in the regulation to make it more efficient, but also to provide the bacteria with substrates through which to synthesize the support compounds that the plant needs [21,22]. In this way, it has been described how some bacteria are capable of inducing the production of lipids and polysaccharides in the plant, as an element related to the growth process and the defensive system [23,24]. In addition to influencing genetic regulation in the plant, certain compounds provided by bacteria could improve the availability of ions, such as iron or calcium, as well as their distribution throughout the plant (improvement in ion transfer) [25,26]. These ions, which are essential for plant growth, in turn have a cascading influence on the production of metabolites related to plant productivity and development. In this way, deciphering the metabolic regulation in plants behind the beneficial interactions with bacteria, becomes a relevant feature to control and improve such interactions.

Among the best studied genera of bacteria beneficially interacting with plants, *Bacillus* has been frequently reported [27,28]. This genus includes a huge variety of species and subspecies that have been cataloged as beneficial in different fields, some being multispectral in their interaction. In general, this group of bacteria have in common a broad genetic background, which makes them very interesting in many agricultural aspects. Among them, the motility and aggregation capacity, the production of phytohormones, the ROS-scavenging and detoxifying actions, or the resistance to stress are quite common skills. In particular, the case of *Priestria megaterium* (formerly *Bacillus megaterium*) is a very interesting strain in agriculture: it is an ubiquitous soil strain with the ability to promote the growth in a number of different plants, and with high resistance to stress and biocontrol effect [29,30,31,32,33]. In addition, the recruitment process of the *P. megaterium* YC4-R4 strain by the plant under stressful conditions and in the context of the epigenetic regulation was recently accurately described [34]. Its beneficial interaction process consists mainly of the production of phytohormones, the solubilization of nutrients, and the production of protective/compatible compounds to enhance plant responses to stress. It is, therefore, a very interesting strain to use as a model of interaction and to study a future application in agriculture. In the present study, we make use of massive sequencing techniques and biochemical assays to understand the metabolic regulation in the plant behind the interaction with this beneficial strain. This will allow us to better understand the aspects necessary to use this strain as a biofertilizer.

## 2. Materials and Methods

### 2.1. Bacteria Growth and Inoculation Process

In this work, we used *Priestia megaterium* YCR-R4 as plant growth-promoting test strain. This strain has been characterized as a phosphate solubilizer, siderophore, and indoleacetic acid producer, as well as able to regulate ethylene by producing aminocyclopropane carboxylate (ACC) deaminase; moreover, this strain was able to induce the IRT1 gene in *Arabidopsis thaliana*, one of the main markers of plant growth induction [34]. This strain was cultured on Luria–Bertani (LB) agar plates and refreshed monthly. The bacterial inoculum for the tests was prepared by dispersing a single colony in 5 mL of LB liquid medium and incubating it for 10 h (30 °C; 150 r.p.m.). Then, this fresh culture was scaled in the necessary volume of LB to inoculate 40 mL per pot. These new cultures were incubated under same conditions until reaching an optical density of OD_600nm_ ≈ 1.0, for an approximate amount of 10^8^–10^9^ colony-forming units (CFUs) per mL. After centrifugation at room temperature for 10 min. at 7000 r.p.m., the bacteria were resuspended in same volume of 0.45% sodium chloride saline solution (as a carrier for inoculation), in order to maintain the OD of the culture. 

### 2.2. Plant Material, Growth Conditions, and Phenotyping

For this test, we used *Arabidopsis thaliana* ecotype Col-0 as model plants. Fresh seeds were surface-sterilized sequentially by using 100% ethanol (1 min) and 20% sodium hypochlorite solution (15 min), washing three times with sterile double-distilled water at the end of the process. The seeds were then plated on ½ × MS (Murashigue & Skoog) agar (0.7%) plates and stratified for 48 h at 4 °C. Then, they were placed in a growth chamber (Femor Gewächshaus, Germany; 22 °C; 12 h of the day–night light cycle, 60% relative humidity, and about 170 μmol/m^2^·s light intensity) to germinate and grow for 5 days before being transferred to the soil. The soil mixture was composed of turf:vermiculite (3:1, *v*/*v*) mix (SIRO Turfa (Mira, Portugal) and PROJAR Vermiculite 0,5-3 mm (Agualva-Cacém, Portugal), respectively), and 50 mL pots were fulfilled with this mix. The seedlings (5 per pot) were kept covered with transparent film for 24 h to maintain soil moisture and improve their adaptation and survival before inoculating them. Inoculation was carried out with 40 mL of the previously mentioned 0.45% NaCl carrier solution per pot, and carefully dispensed around each plant. Five pots and three repetitions of each treatment were carried out. Nine days after treatment (DAT), the phenotype of the seedlings was recorded by root length and total dry weight (DW) biomass determination. Samples for subsequent tests were prepared from fresh material, dry-grinded material, or grinded to powder by freezing with liquid nitrogen and mortaring, depending on each test requirement.

### 2.3. Total RNA Extraction and Transcriptomic Analysis

Three biological replicates of whole-seedling samples were prepared 7 days after treatment (when the phenotype was still not noticeable). They were frozen with liquid nitrogen to avoid degradation and ground with a mortar. RNA was extracted with the RNeasy Plant Mini Kit (Qiagen), following the manufacturer indications. Thereafter, library preparing (NEBNext Ultra Directional RNA Library Prep Kit for Illumina; New England Biolabs, Ipswich, MA, USA) was carried out with 1 μg total RNA per sample. Hence, the library and sequencing were prepared by the Core Facility for Genomics at the Shanghai Center for Plant Stress Biology. Subsequently, the raw data were pre-processed with Trimmomatic (v0.36) and clean reads were analyzed with Rockhopper (v2.04) in order to ensure the quality of the data, followed by HISAT2 (v2.1.0) for clean read alignment (using TAIR10 as reference), and HTSeq-count (v0.9.1) for the read count matrix. Finally, to identify DEGs, we used edgeR software, adjusting the cut-off fold-change ≥ 1.5 and false-discovery rate ≤ 0.05. The data discussed in this publication have been deposited in NCBI’s Gene Expression Omnibus [35] and are accessible through GEO Series accession number GSE199501 (https://www.ncbi.nlm.nih.gov/geo/query/acc.cgi?acc=GSE199501 (accessed 30 March 2022).

### 2.4. Gene Expression Level by qPCR

The quantification of key genes involved in plant–bacteria interaction was initially performed by using TransScript One-Step gDNA Removal and cDNA Synthesis Super Mix (TransGen Biotech, Beijing, China) to synthesize cDNA from RNA samples. Subsequently, quantitative PCR was performed using iTaq Universal SYBR Green Supermix (Bio-Rad, Hercules, CA, USA) in a CFX96Touch Real-Time PCR Detection System (Bio-Rad). Here, the program used consisted of an initial denaturation at 95 °C (3 min), a 40-cycle amplification phase, a denaturation phase at 95 °C (10 s), and finally, an annealing phase at 60 °C (45 s). As internal control, the housekeeping gene actin 2 (AT3G18780; ACT2) was included [36]. The primer list is included in Table 1.

### 2.5. Iron Determination

For the iron content in the seedlings, we followed up the colorimetric assay proposed by Gautam et al. [37]. In brief, 9 DAT plant samples (200 mg) were completely dried for 2 days and then ground to powder. Once mineralized with nitric acid (65%) and H_2_O_2_ (30%) solutions, 5 μL of each sample was dissolved in 1 mM bathophenanthroline disulfonic acid solution (containing 0.6 M sodium acetate and 0.48 M hydroxylamine hydrochloride) up to 250 μL, and incubated at room temperature for 5 min. Iron presence was revealed by pink/red coloration in the solution, and it was measured at 535 nm in a spectrophotometer (Multiskan Sky, ThemoFisher Scientific, Waltham, MA, USA). Iron standards were used for the calibration curve. This experiment was repeated three times with similar results and n = 3 biologically independent samples.

### 2.6. Quantification of Chlorophyll Levels

The chlorophyll content was measured in 3 leaves per plant (6 plants in total) at 9 DAT. Pigment extraction was performed by submerging a disk of 0.3 cm per leave in 10 mL of 100% methanol for 24 h, in darkness. After incubation, the samples were centrifuged at 5000 rpm for 10 min at room temperature. The disks were previously weighed (fresh weight) to normalize the data, and the chlorophyll and pigments content (C_a_, chlorophyl a; C_b_, chlorophyl b; and C_x+c_, xanthophyles and carotenoids) of the supernatants was measured by reading optical density (OD) using a Multiskan Sky spectrophotometer (ThemoFisher Scientific, USA). To quantify the chlorophyll content of the samples, we measured the optical densities at the wavelength of 666, 653, and 470 nm, and applied Wellburn’s equations to determine pigments’ concentration [38] by using the following formulas:C_a_ = 16.72 *A*_666_ − 9.16 *A*_653_
C_b_ = 34.09 *A*_653_ − 15.28 *A*_666_
C_x+c_ = (1000 *A*_470_ − 1.63C_a_ − 104.96C_b_)/221

This experiment was repeated three times with similar results and n = 3 biologically independent samples.

### 2.7. Cellulose and Lipid Determination

The cellulose content in plants was measured according to Updegraff, 1969, with input from Rui and Anderson [39,40]. In summary, 24 h before sampling, 9 DAT plant samples were placed in darkness and then 100 mg of rosette leaves was incubated overnight in 1 mL of ethanol 80% at 65 °C. Subsequently, ethanol was removed and the samples were incubated overnight in 1 mL of acetone at room temperature. After the acetone, dried samples were completely ground and the resulting powder was weighed for normalization thereafter. Subsequently, 1 mL of acetic acid:nitric acid:water (8:2:1) solution was added and boiled 30 min. After centrifuging the mixture at 10,000 rpm for 5 min, the pellets were resuspended in 1 mL of sulfuric acid solution (67%). Finally, a 50 μL aliquot was added to a tube containing 1 mL of 0.2% anthrone solution in sulfuric acid. After 10 min of incubation at room temperature, the samples were measured at 620 nm on a spectrophotometer. Pure cellulose was used as standard.

On the other hand, three replicas of 9 DAT plant samples were prepared to determine the lipid content by the phospho-vanillin reagent method, following the indications in Men et al. [41], with slight modifications. Briefly, 100 mg of plant was homogenized in 100 μL of sodium sulfate solution (2%), and completed up to 1 mL with chloroform: methanol solution (1:1). After centrifuging at 10,000 r.p.m. for 5 min, the supernatant was collected and mixed with 300 μL of distilled water. The mixture was then centrifuged under the same conditions, and the chloroform layer was transferred to a 96-well microplate. After the sample was completely dried, 50 μL of 98% sulfuric acid was added, and the plate was incubated for 20 min at 90 °C. Finally, 150 μL of phospho-vanillin reagent was added. Absorbance was measured at 530 nm after cooling for 10 min. A triolein solution was used as standard. Both experiments were repeated three times with similar results and n = 3 biologically independent samples.

### 2.8. Determination of Total Phenol and Flavonoid Compounds

To determine total phenol content, we used the Folin–Ciocalteu (F–C) reagent derived from Ainsworth and Gillespie [42], with minor modifications from López-Hidalgo et al. [43]. Briefly, 95% methanol was used to extract 50 mg of freeze-ground material, with the help of ultrasound treatment for 30 min. The sample was centrifuged at 10,000 r.p.m. for 15 min and the extract was decanted. The supernatant was later mixed and diluted with methanol. Thereafter, 200 μL of 10% F–C reagent was added to 100 μL each sample and incubated at room temperature for 2 min. Finally, 800 μL of 700 mM sodium carbonate was added and incubated in darkness for 2 h. The samples were measured in a 96-well microplate at 720 nm. The total phenol content was expressed as gallic acid equivalents using a calibration curve. 

On the other hand, the total content of flavonoids was determined by using the aluminum chloride method [44]. Briefly, 150 μL of supernatant, obtained in the previous process, was mixed with 20 μL of 10% (*w*/*v*) aluminum chloride and 20 μL of 1 M potassium acetate. The mixture was diluted with 700 μL of 95% methanol and incubated at room temperature for 30 min. Samples were transferred to a 96-well microplate and the absorbance was measured at 415 nm. The total content was expressed as quercetin equivalents using the calibration curve. Both experiments were repeated three times with similar results and n = 3 biologically independent samples.

### 2.9. Soluble Sugar Measuring 

To ensure the content of soluble sugar, the anthrone method [45], with minor modifications, of López-Hidalgo et al. [43] was used. In brief, 50 μL of supernatant (from the previous procedure) was added to 750 μL of anthrone reagent and incubated at 100 °C for 10 min, and then cooled on ice for 10 min. Then, the sample was transferred to a 96-well microplate and measured at 625 nm. The content was calculated with respect to a d-glucose in a standard curve of ethanol 80%. 

### 2.10. Statistical and Analysis Software

Statistical analysis was performed in Prism (v5.04; GraphPad Software). Therefore, we applied Student’s *t*-test and one-way ANOVA (with Tukey’s post-test) for pairwise and multi-group comparisons, respectively. The significancy level was set up at *p* ≤ 0.05. In transcriptome analysis (Gene Ontology enrichment), we used Cytoscape (v3.9) software, implemented with the BiNGO plug-in (default configuration database for *Arabidopsis thaliana*).

## 3. Results

### 3.1. Bacteria Inoculation and Plant Phenotyping

Nine days after carrying out the inoculation process, the early phenotype began to be visible, as shown in Figure 1a. At that point, the main root length was measured, as well as seedlings’ total dry weight (DW) under each condition. The seedlings inoculated with *Priestia megaterium* YC4-R4 (Treated) showed bigger leaf appearance (Figure 1a); meanwhile, the main root length reached up to 30% higher values (4.31 ± 0.19 cm, compared to 3.23 ± 0.33 cm in control plants) (Figure 1b; Appendix A). Similarly, the dry weight achieved by the inoculated plants was almost double (20.8 ± 1.15 mg) compared to that registered in the non-inoculated ones (10.96 ± 0.87 mg) (Figure 1c).

### 3.2. Total RNA Extraction and Transcriptomic Analysis

In order to unveil insights into plant responses to the treatment with *P. megaterium* YC4-R4, we performed RNA sequencing analysis (RNA-Seq) to investigate the transcriptomes of treated versus untreated plants. Thus, 7 DAT resulted in 734 upregulated and 630 downregulated differentially expressed genes (DEGs) (fold-changes ≥ 1.5; false-discovery rate ≤ 0.05), with respect to control conditions. Gene ontology (GO) analyses showed that growth-and-development-related DEGs were in general upregulated after bacteria treatment (Figure 2). Meanwhile, the downregulated DEGs were highlighted by the three categories including defense responses, ethylene biosynthesis, and regulation, as well as biosynthesis of indole, phytoalexin, and camalexin (downregulated GO analysis in Appendix A; heatmap in Appendix A). 

Among upregulated GO, the main regulated areas include lipid biosynthesis, cell wall biogenesis, carbohydrate biosynthesis, phenol and flavonoid biosynthesis, and metallic and organic ion transport. They suggest that the recorded phenotype was obtained through the induction of DEGs related to the biogenesis and reorganization of the cell wall, such as glucans, cellulose, polysaccharides, lipids, phospholipids, or glycerolipids biosynthesis. Similarly, another area of interest in plant growth and development was notably represented by DEGs related to ion transportation. We found upregulated DEGs of organic ion transport of some of the most relevant nutrients, such as phosphate or nitrogen (as ammonium transporters), but also metallic ion transport, such as molybdenum, zinc, or ion. These last two types of ion transporters are very relevant in growth and development by cell-wall-related enzymes and hormones (i.e., metalloenzymes and indoleacetic acid) and in chlorophyll biosynthesis. Some other growth-related GOs were upregulated as well, as with amino acid metabolism, glucosinolate biosynthesis, or sulfate assimilation. Finally, reactive oxygen species scavengers such as phenols or flavonoids were also detected as part of the facilitation of growth promotion by the bacteria treatment. 

On the other hand, downregulated DGEs were mainly corresponding to GOs of defense regulation, such as the general immune response, jasmonic acid response, and ethylene response and signaling. These GOs are related to the hosting process of beneficial interaction with microbes, which indicates the positive response of the plant to the treatment with *P. megaterium* YC4-R4. All DEGs considered under this analysis could be found in the Appendix A.

### 3.3. Gene Expression Level by qRT-PCR

In order to assess transcriptomic analysis results, we decided to the check gene expression level of some key genes of each area of relevance highlighted by the RNA-seq study, including IRT1 and FRO2 (which regulate iron transportation and are related to plant growth), XTH14 (xyloglucan endotransglycosylase and in relation to cell wall biogenesis), FLS1 (flavonol synthase), SPS4F (sucrose biosynthesis) and LAS1 (lanosterol synthase) (Figure 3). Consequently, the quantitative real-time PCR results further supported previously described gene ontology analyses. In this way, the upregulation of two of the main iron transport genes showed how the inoculation with *P. megaterium* YC4-R4 was able to induce the plant growth by enhancing the transportation of metallic ions as Fe^2+^, necessary for chlorophyll synthesis. Similarly, the upregulation of some key genes for lipid and carbohydrate biosynthesis may be related to plant growth regulation in terms of cell wall biogenesis. In the case of the XTH14 gene, it was even induced by 4-fold. On the other hand, phenols’ and derivates’ biosynthesis were shown to be as well upregulated as part of the plant growth-promoting mechanisms, especially related to reactive-oxygen-species scavenging during such a process. These results collectively suggest that the bacteria-induced transcript regulation plays an important role in plant general growth patterns. In Appendix A, we included the gene expression level for some of the downregulated key gene expression levels.

### 3.4. Iron and Chlorophyll Content

The amount of iron and total chlorophylls was recorded 9 DAT, coinciding with the early emergence of the growth promotion phenotype induced by treatment with *P. megaterium* YC4-R4. Thus, the amount of iron accumulated in the treated seedlings was up to 4% higher (483.3 ± 10.6 μg·g^−1^ DW) than that accumulated in those of the control group (467.1 ± 18.8 μg·g^−1^ DW) (Figure 4a). Regarding total chlorophyll content, despite observing a slight increase in it, this was not significant in the treated seedlings with respect to the control group (Figure 4b). In this early phenotype stage, it is shown that a greater accumulation of iron is sufficient to produce more biomass in the treated seedlings, without even showing a significant increase in their content of chlorophylls. 

### 3.5. Cellulose and Lipid Determination

After observing an increase in total biomass in plants with bacterial treatment, the metabolomic factors behind it were evaluated. Thus, the measurement of the total content of cellulose and lipids was carried out as the most relevant compounds for the formation of new cell walls. Thus, the seedlings treated with *P. megaterium* YC4-R4 showed an increase of slightly higher than 7% (118.3 ± 6.7 μg·g^−1^ FW) compared to the seedlings of the control group (109.7 ± 6.6 μg·g^−1^ DW) (Figure 4c). On the other hand, the total % of lipids recorded was also higher in the treated seedlings (almost 38%) compared to those in the control group (35%) (Figure 4d). The higher accumulation of cellulose and the higher percentage of lipids in the processed samples of treated seedlings are shown as a direct support of the early phenotype of higher DW detected above. 

### 3.6. Total Phenol and Flavonoid Content

The presence of phenols and flavonoids is usually related to plant defense, as well as part of the antioxidant systems of plants. In general, growth and development processes produce reactive oxygen species (ROS), which are increased in proportion to the increase in biomass generated. Thus, we carried out the determination of the total phenols and flavonoids of the treated seedlings compared to those of the control group. Starting with the determination of total phenols, the seedlings treated with *P. megaterium* YC4-R4 showed an increase of about 45% (16.243 ± 0.576 μg·g^−1^ FW) compared to the seedlings of the control group (9.118 ± 1.144 μg·g^−1^ DW) (Figure 4e). Regarding the total flavonoids, we also detected an increase in their production in the treated seedlings (3.112 ± 0.346 μg·g^−1^ FW), compared to those of the control group (2.689 ± 0.188 μg·g^−1^ FW), assuming an increase in accumulated flavonoids of 13.6% (Figure 4f). 

### 3.7. Soluble Sugar Content

Finally, considering that the synthesis of simple sugars and compounds derived from carbohydrates is directly related to the plant’s growth rate, we decided to determine the total soluble sugar content after the treatment with *P. megaterium* YC4-R4. Thus, measurements of soluble sugars showed that the treated seedlings accumulated only 2% more (151.01 ± 6.58 μg·g^−1^ FW) compared to the seedlings of the control group (147.79 ± 6.59 μg·g^−1^ DW), not enough to be statistically significant (Figure 4g). Probably, the rate of consumption of the same to carry out the vegetative growth leads to the accumulation of the same in treated plants not showing to be significant. This experiment was repeated three times with similar results and n = 3 biologically independent samples.

**Figure 4 plants-11-03039-f004:**
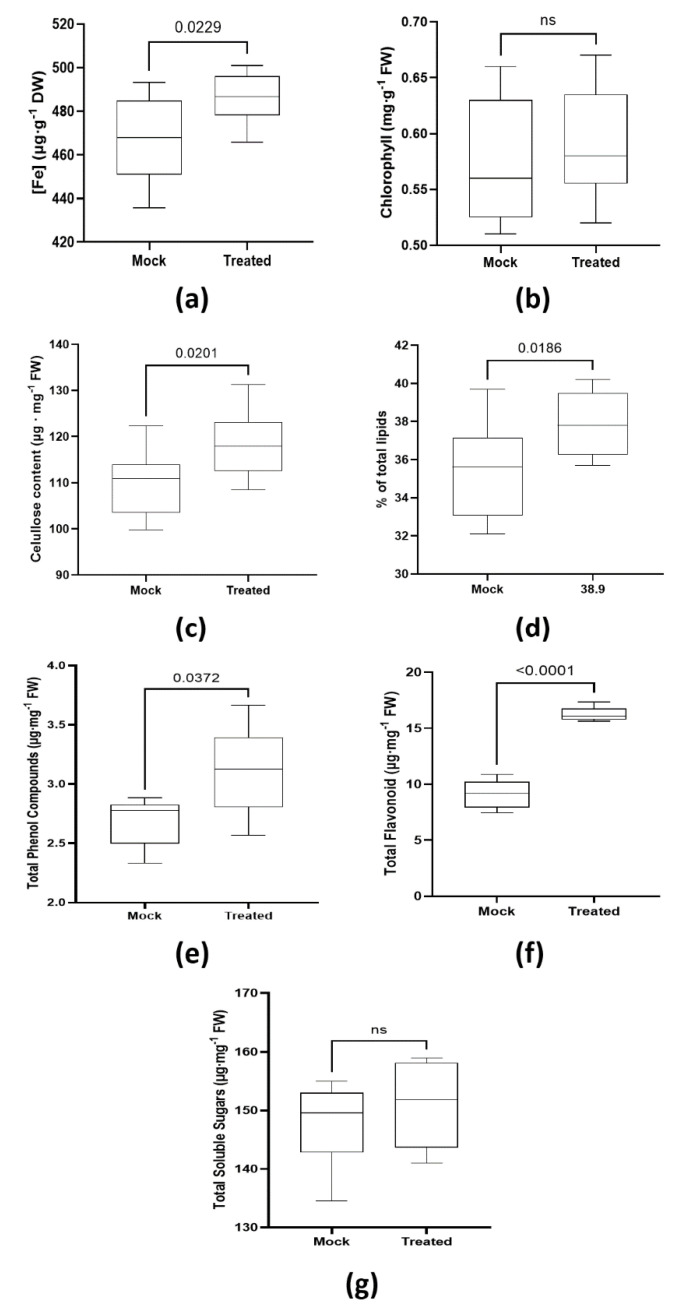
Plant-growth-promotion key metabolite determination. The panel shows the determination of different compounds related to the bacteria-mediated plant growth-promotion process. Thus, (**a**–**f**) boxplots show the iron content (**a**) and the total chlorophyll content (**b**); the total cellulose (**c**) and lipid proportion (**d**); total phenol (**e**) and flavonol content (**f**); and the total soluble sugar content (**g**). Three repetitions of three replicas (n = 9) were prepared for this test. All the boxplots and violin plots in this figure show representative data from three independent experiments (each one with n = 9 biologically independent samples). Whiskers represent the minimum to maximum data range, and the median is represented by the central horizontal line. The upper and lower limits of the box outline represent the first and third quartiles. Dots in violin plots represent the dispersion of each sample recorded. Error bars represent s.d., including *p*-value when they were significant, or ‘ns’, when they were not significant.

## 4. Discussion

Nowadays, bacterial biofertilizers are considered to represent a sustainable alternative with great potential. Their combined use may drastically reduce the use of phytosanitary products of chemical origin, also reducing their impact on the soil and the environment [16,17,46]. However, these products also require a good understanding of their working mechanisms, not only to enhance their use and make them more efficient, but also to guarantee their safety and reliability in the future [47,48,49]. In this way, they have usually been classified as having a direct action, for example, improving the availability of nutrients, or an indirect action, such as the biocontrol of pathogens [16]. However, we have to consider that the involved mechanisms and effects can go further. 

In this work, we have sought to elucidate how the regulation of the central and secondary metabolism of the plant occurs during the processes of beneficial interaction, through a transcriptomic and biochemical analysis of some of the most relevant development-related compounds. The model strain, *Priestia megaterium* YCR-R4, as with many other related to *Priestia* or *Bacillus* genus, has been described as a beneficial strain capable of solubilizing phosphates, producing auxins, or controlling ethylene levels via aminocyclopropane-1-carboxylic acid (ACC) deaminase [30]. Moreover, similar kinds of strains have been reported as well to be able to modulate some specific mechanisms in plants [50,51]. By using *Arabidopsis thaliana* Col-0 as a model plant, we detected, firstly through transcriptomic analysis and gene expression level, that many central metabolism-related genes were upregulated. These results were assessed by biochemical tests, showing this regulation was finally reflected in an increase in total cellulose and in percentage of lipids, which is consistent with previous studies regarding the accumulation of these compounds in plants with accelerated growth rates, capable of tolerating environmental stresses, as well as resistant to pathogen infections [52,53,54,55,56]. In this sense, some authors such as Pršić and Ongena have reported how beneficial bacteria enhance the cell wall related to plant immunity triggering [57]. In our study, defensive systems were downregulated in plants under treatment, indicating the cell wall biogenesis was related to a direct vegetative growth instead of a defensive mechanism. Moreover, some secondary metabolism biosynthetic pathways were upregulated as well. One of the most interesting was the increasing levels of phenols and flavonoids. Despite them being described as ROS scavengers (which can over-accumulate in growth processes and limit them) or defense activation molecules, many studies have pointed their influence on the bacteria-mediated plant growth promotion process [58,59,60]. In this sense, these compounds have also been described as fundamental as signal molecules in cell wall biogenesis [61,62,63,64].

Finally, it should be noted that, through transcriptomic analysis, we were able to evaluate the positive regulation of other genes involved in cell wall organization, such as isoprenoid (dolichol), phospholipids, or glycolipid biosynthesis-related genes. Some of them have been already reported as genes induced under bacteria interaction processes [65,66,67]. In addition, the upregulation in the synthesis of sugars and polysaccharides (cellulose, glycoside, sucrose, and glucosinolate) was also revealed after transcriptomic analysis. Although these compounds did not notably increase in the treated samples (at least the soluble ones), these compounds may be rapidly consumed, so the detection technique could not be resolutive enough. Alternatively, they may be employed in fast energy consumption to meet the needs of accelerated growth [68]. Probably, this rate of consumption in vegetative growth leads to the treated plants not showing a significative accumulation of sugars [69,70].

Furthermore, through transcriptomic analysis, we were also able to observe an upregulation of other genes directly related to plant nutrition and the production of essential compounds such as amino acid derivatives. Among these genes, those related to the transport of ammonium ions and phosphates, as well as the regulation of the assimilation of sulfates, stand out, being considered as especially relevant for plant nutrition and development. On the other hand, considering the amino acid derivatives’ upregulation, the synthesis of nicotianamine is noticeable as a metal-chelating agent that can supplement the siderophores produced by the bacterium *Priestia megaterium* YC4-R4 to guarantee the necessary amount of metal ions (as Fe^2+^ and Zn^+^) required for plant vegetative growth [30,71,72]. Likewise, through the biochemical study, we were able to describe a substantial increase in iron concentration in the plant. However, this did not significantly translate into an increase in total chlorophyll. This can be explained because the content per milligram of tissue remains the same, but the plant generates more tissues, as it needs more chlorophyll and iron [73]. Along the same lines, some authors have described that a larger plant size is not always linked to a higher chlorophyll content. This is the case described for larger genotypes of *Artemisia cina*, shown by Kasmiyati, Kristiani and Herawati, where the total amount of chlorophyll was equal to or slightly less than that of smaller genotypes [74].

Regarding plant general growth, on some occasions it has been mentioned that, despite being treated with a biofertilizer, this is not fully reflected in an increase in fruit production [75]. Despite nutrients as nitrogen being more accessible, a higher vegetative growth was not linked to higher productivity level [75]. This induced growth in certain tissues, such as roots or stems, becomes highly relevant since it is likely to result in a better state of health or a better response to mild environmental stresses, linked to an efficient regulation of phytohormones, but still not enough to consider a better final production [76,77,78,79,80]. Further tests considering yield are necessary to cope with this question. However, in the case of some forage crops such as alfalfa (*Medicago sativa*), it could be especially interesting to simply stimulate vegetative growth per se, since the use of these plants mainly includes as forage the vegetative tissues such as leaves and stems [81,82,83]. Nevertheless, in the case of beneficial interaction with microorganisms, we can recommend a higher cell wall production and plant size as factors to assess the general health status of crops treated with beneficial bacteria and other types of biofertilizers.

## 5. Conclusions

In this study, we described how a plant growth-promoting model strain, *Priestia megaterium* YC4-R4, is capable of influencing the central and secondary metabolism of the plant to increase its growth. In this way, we were able to verify through transcriptomics, gene expression, and biochemical determinations how this strain promotes basic growth areas such as cell wall biogenesis or metal ion transport. In addition, *P. megaterium* YC4-R4 was able to induce the production of phenols and flavonoids, which in this context, may be acting as positive growth regulators, signal molecules for development, as well as for scavenging ROS molecules. In addition, the positive regulation for the use of basic nutrients and sugars, as well as the negative regulation of the plant’s defensive response, has also been supported in this work. Hence, this is one of the first plant–microorganism models in which we have been able to assess that the promotion of growth is not only given directly due to nutrient supplementing, but also by influencing the central and secondary metabolism in order to increase the rate of vegetative growth and improve the efficiency of the associated mechanisms.

## Figures and Tables

**Figure 1 plants-11-03039-f001:**
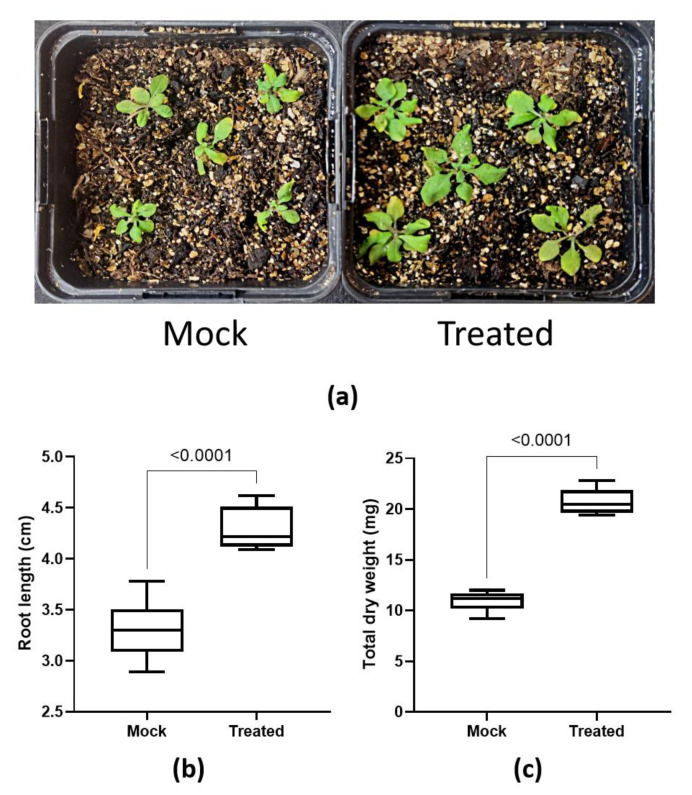
Plant growth promotion test. (**a**) The picture shows the 9 DAT phenotype of seedlings under ‘Mock’ (control) and ‘Treated’ conditions; (**b**,**c**) boxplot graphs record the main root length (**b**) and dry weight of the whole plant (**c**). Four independent experiments (each one with n = 5 biologically independent samples) were prepared in this test. Whiskers represent the minimum to maximum data range, and the median is represented by the central horizontal line. The upper and lower limits of the box outline represent the first and third quartiles. Error bars represent s.d., including *p*-value when they were significant.

**Figure 2 plants-11-03039-f002:**
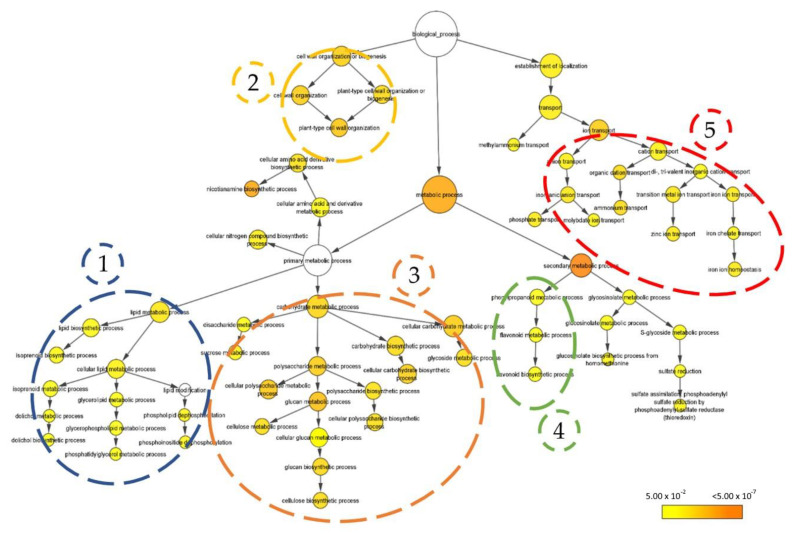
Plant transcriptomic regulation under inoculation with *P. megaterium* YC4-R4. Gene Ontology categorization of YC4-R4-induced DEGs. The size of solid-lined circles represents the number of genes in each category; meanwhile, the color indicates the significance of gene expression in each category. Clustered biological processes are indicated manually by outline-dashed circles, numbered as (1) in blue, for lipid biosynthesis; (2) in yellow, for cell wall biogenesis; (3) in orange, for carbohydrate biosynthesis; (4) in green, for phenol and flavonoid biosynthesis; and (5) in red, for metallic and organic ion transport. See Appendix A.

**Figure 3 plants-11-03039-f003:**
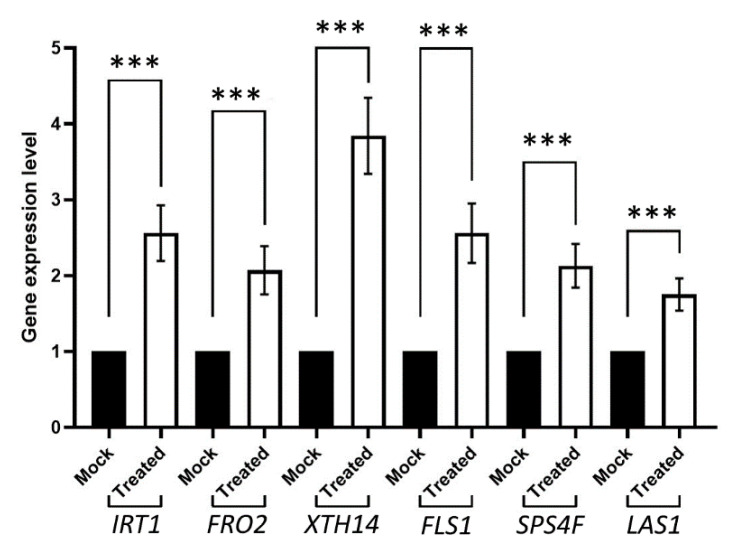
Relative expression levels of key genes. The bar graph shows representative quantitative real-time PCR results from three independent experiments (each one with n = 3 biologically independent samples) for key genes in the areas of interest highlighted by transcriptomic analysis. Asterisks (***) stand for *p*-values > 0.001. Error bars represent s.d.

**Table 1 plants-11-03039-t001:** Primers used in qRT-PCR procedure for gene expression level evaluation.

Gene	Locus	Read	Sequence (5′–3′)
*IRT1*	AT4G19690	Forward	GGAAGAATGTGGAAGCGAGT
		Reverse	TCTGGTTGGAGGAACGAAAC
*FRO2*	AT1G01580	Forward	ATAGGGAGACGAAGGGAGGA
		Reverse	AGGAGTGATAGTGGCGAAGC
*FLS1*	AT5G08640	Forward	TCCTCACTTCCTCCCTCCTT
		Reverse	CGCTGGTTGTTCTTTCTCTG
*XTH14*	AT4G25820	Forward	CATCCTTACACTATCCACACCAA
		Reverse	CCACCCCATTTTTCTCGTT
*SPS4F*	AT4G10120	Forward	GCTCTTTGTGGTTGCTGTTG
		Reverse	CGCTTTGATGTTTCCGTTGT
*LAS1*	AT3G45130	Forward	TGTTTCTCTTGCCTGCTCTG
		Reverse	GTAGTCCCCATCCTCCATCC

## Data Availability

Not applicable.

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
