# Peer review of "Discerning Transcriptomic and Biochemical Responses of Arabidopsis thaliana Treated with the Biofertilizer Strain Priestia megaterium YC4-R4: Boosting Plant Central and Secondary Metabolism"

_plants, 2022, doi:10.3390/plants11223039_

Round 1

Reviewer 1 Report

The submitted manuscript is a good study. I have no main critical comments concerning the author conception, results, their presentation and discussion. In my opinion, this research may be published after minor revisions (related to the style of some sentences and addition of some information), which could be easily make by authors.

Please consider some specific comments and suggestions listed below.

Line 22 – Change "analysis" to "analyses"

Line 31 – Delete an excessive comma between "conditions" and "are"

In my opinion, some sentences could be simplified to make them more facilitate and understandable for readers. 

Some examples are below:

Line 32 "In soil first challenge, the health…" – the phrase is difficult to comprehend

Lines 39-42 could be rewritten. E.g. line 39 "…an alternative to  better crop management…"  I would suggest the following: In this context, the use of beneficial bacteria has been shown to be a promising way to manage crop better. Line 40  – "Its uses have been fertilization…"  It would be probably better to put this more straightforward here: "They were used as/for…" For instance, "They have been applied for fertilization and improvement of plant growth (biofertilizers), protection against pests and pathogens (biocontrol), or improvement of the response to stress in plants [8-10] ".

Lines 48. What does "widespread use" mean here? Do you mean other widespread crop management, e.g., chemical control and fertilization or using the bacteria in some widespread way not mentioned above (see lines 40-42)? Please, clarify. 

Line 85 – "…it is a ubiquitous soil strain, with multiple strains" Strain with many strains? Did you mean that it (P. megaterium) is a ubiquitous soil bacterium / soil-born species with multiple strains?

Lines 101-102 – Please check and clarify the sentence "The bacterial inoculum for the tests was prepared by growing a single colony of each strain was dispersed in 5 mL of LB liquid medium for 10 h (30°C; 150 r.p.m.) ". Why "…single colony of EACH strain"? You used YCR-R4 only. Was the inoculum prepared by growing a single colony followed by dispersion of bacterial cells in 5 ml of LB…?

Lines 104 & 107 – the same

Line 113 – What was used for seed treatments under these conditions?

Line 124 – Why 7 days after the treatment were chosen for RNA-Seq?

Lines 177 – "After centrifuge…"  Perhaps, "After centrifugation of…" Line 186 – the same suggestion.

Line 181, 191 and sections 2.9 and 2.10 – It would be desirable to provide an information on the standards used (company, country). As far as I remember, it is a conventional rule of MDPI-journals.

Line 246 Why folds-change>1.5 was used as a threshold for the considered DEGs?

Line 270  – While decrease of the plant immune response may be considered as “the positive response of the plant to the treatment” in terms of welcoming of a beneficial microbe, it may be not so good for a plant itself. If the suppression of plant defensive systems has a long-standing effect, it may decline the overall resistance to pathogens devaluating the fertilizing benefit of the treatment. It would be nice to comment a little this contradiction in the Discussion. Such a discussion point would fit well with your absolutely correct reasoning about the necessity of good understanding of mechanisms of action and guaranteeing the safety of bacteria’ used.

Line 271In the attached Supplementary table 1 only the upregulated genes are mentioned. Please add the list of downregulated DEGs as well.

Lines 312-313, 324-325, 338-339, and 348-349 – To avoid monotonous repeating, I would recommend moving these sentences to the section 2.11, mentioning that all experiments involving biochemical analyzes were repeated three times with similar results (n = 3 biologically independent samples in each experiment).

Reviewer 2 Report

Santos et al. provided a Research article on “Discerning transcriptomic and biochemical responses of Arabidopsis thaliana treated with the biofertilizer strain Priestia 3 megaterium YC4-R4: boosting plant central and secondary metabolism”. Though the topic is of current interest, the manuscript needs a substantial revision to be published. I have mentioned a few points of the manuscript that should be considered by the authors, before publishing.

·         Provide a good resolution image for fig.1a.

·         Provide picture showing roots of seedlings.

·         In fig.2 use Venn diagram and Heat map to simplify the transcriptomic analysis.

·         In materials methods mention the formula used in the measurement of chlorophyll content.

·         You are suggested to check the crop yield also like fruit production of the plant because in case of nitrogen nutrient higher vegetative growth is not linked with higher productivity.  

·         Recheck all the references.

·         In figure 1- Phenotypic data, Leaf size and root length measurement actual value are not given only given 30% increase in the treated plants as compare to control but in dry weight content, actual weight of treated and control plants both are given, so give actual leaf size and root length of control and treated plants.

·         In gene expression analysis (qRT-PCR)- only Up regulated genes were analyzed in transcriptome analysis of treated plants as compare to mock, so also analyze the down regulated gene involved in  Pathogenesis, Signaling, and other key regulatory processes.

·         Give data for final yield of treated plants as compare to control plants.

·         In line 131, the use of Trimmomatic v0.32 is not well mentioned (Trimmomatic is used to trim and filter low-quality reads).

·         This study does not mention which tool is used to validate the quality of the filtered reads (like FastQC, fastp etc.).

Author Response

Dear reviewer,

After your helpful comments, please find here the new version for this work. We have followed the tracking indications as well as solved all the requested comments. Please, find here below the enumeration of main changes applied considering the comments we have received:

- Provide a good resolution image for fig.1a.

A: Thanks for the comment. Maybe the size of the inserted figure seems not the best quality, but the figure apported in TIFF format is high quality. We have included a better resolution in the inserted (in manuscript text) to enhance the quality as well.

- Provide picture showing roots of seedlings.

A: Thanks for the suggestion. Please find it as Supplementary Figure 1

- In fig.2 use Venn diagram and Heat map to simplify the transcriptomic analysis.

A: Thanks for the suggestion. We don’t find the Venn diagram is helping in this case. The heatmap was performed, but we consider better as a Supplementary figure to support the GO analysis graph. Also, heatmap for downregulated DEG was included as supplementary material.

- In materials methods, mention the formula used in the measurement of chlorophyll content.

A: The formulas were the one optimized for methanol. Now they are indicated in the text. Thanks for the suggestion.

- You are suggested to check the crop yield also like fruit production of the plant because in case of nitrogen nutrient higher vegetative growth is not linked with higher productivity.  

A: We highly value your suggestion. However, at this point, we cannot reproduce this test again. We will consider for upcoming test as a relevant point to take into the account. By the moment, we revise all the nitrogen-related part of the work to not assume higher vegetative growth is linked to higher productivity. Thanks!

- Recheck all the references.

A: So sorry, probably a general mistake during curating phase of the bibliography. In previous stages we didn’t detect this. Now all references are correct.

- In figure 1- Phenotypic data, Leaf size and root length measurement actual value are not given only given 30% increase in the treated plants as compare to control but in dry weight content, actual weight of treated and control plants both are given, so give actual leaf size and root length of control and treated plants.

A: Thanks for the comment. Leaf size may curse into confusion. Our meaning was the appearance was bigger, but as the leaves withering very fast, measures were not consistent. We decided only to mention the leaves to point the plants seems bigger, supported by picture. Root length is now included for both, control and treated conditions. 

- In gene expression analysis (qRT-PCR)- only Up regulated genes were analyzed in transcriptome analysis of treated plants as compare to mock, so also analyze the down regulated gene involved in Pathogenesis, Signaling, and other key regulatory processes.

A: We considered not to overcrowd the figure, and the showed here are enough to validate the RNAseq. However, we performed gene expression level for other genes as well, including some of the downregulated. We included them in supplementary figure 4. 

- Give data for final yield of treated plants as compare to control plants.

A: As mentioned above, we are not in position to repeat these tests. We adapted all related presumptions in the text, and will consider for upcoming projects.

- In line 131, the use of Trimmomatic v0.32 is not well mentioned (Trimmomatic is used to trim and filter low-quality reads).

A: You’re right. So sorry. Now is corrected in the text.

- This study does not mention which tool is used to validate the quality of the filtered reads (like FastQC, fastp etc.).

A: Sorry, we forget to mention (better said, our bioinformatics team forget to tell us this info). FastQC and Rockhopper were used to cure the data and ensure the quality. Now is incorporated in the text.

Reviewer 3 Report

Dear authors!

The manuscript is devoted to the actual topic of stimulating the growth and development of plants with the help of microorganisms. However, to improve it, I propose to make some corrections.

1.The English language of the manuscript needs further improvement.

2. Keywords do not give an idea of ​​what the text is talking about, they are merely a repetition of words from the title of the manuscript.

3. In section 2.1. a more detailed characterization of the PGP-properties of the YC4-R4 strain should be written.

4. In section 2.2. it is necessary to describe in more detail the methodology for setting up the experiment: what type of soil was used, what are its physicochemical characteristics, in what quantity it was placed in vessels; how many seedlings were placed in each vessel (Fig. 1a shows 5 plants, but it is possible that some of the seedlings died after being transplanted into the soil); how many repetitions were used; how did bacteria inoculate plants?

5. In the section Materials and methods, after paragraph 2.7 is followed by paragraph 2.9.

6. Section 3.2. It is necessary to write a what do the abbreviations DEG and GO mean.

7. What does the sign "***" and the curved lines above the bars of the diagram in Fig. 3 mean?

8. Line 293. "Some of these genes were even induced by more than 4-fold". What are these genes? In Fig. 3, there is only one such gene, ХTH14.

9. Lines 312-313, 324-325, 338-339, 348-349. "This experiment was repeated three 312 times with similar results and n = 3 biologically independent samples." This phrase should be moved to the materials and methods section when describing the relevant methods.

10. Line 328. "The literature shows ...". Reference should be made to literature data.

11. Line 369-370. "...or an indirect action, such as the biocontrol of pathogens or the production of phytohormones". Why do the authors believe that the production of phytohormones by bacteria has an indirect rather than a direct effect on plants?

12. In the Discussion section, links to the authors' own drawings should be provided.

13. In the list of references, there are many occasions when no journal titles in which articles were published are present (â„–â„– 3-5, 7, 10 16, etc.).

Author Response

Dear reviewer,

After your helpful comments, please find here the new version for this work. We have followed the tracking indications as well as solved all the requested comments. Please, find here below the enumeration of main changes applied considering the comments we have received:

- The English language of the manuscript needs further improvement.

A: Thanks for the comment. Effectively, it needed some work in this, so we dedicated a time to improve it.

- Keywords do not give an idea of ​​what the text is talking about, they are merely a repetition of words from the title of the manuscript.

A: Thanks for the comment. We have changed to include main topics of the work

- In section 2.1. a more detailed characterization of the PGP-properties of the YC4-R4 strain should be written.

A: It’s fair to include some more info, but we didn’t want to crowd this section. However, we consider supplying some more info as the reviewer suggested. Thanks, this makes bacteria role clearer.

- In section 2.2. it is necessary to describe in more detail the methodology for setting up the experiment: what type of soil was used, what are its physicochemical characteristics, in what quantity it was placed in vessels; how many seedlings were placed in each vessel (Fig. 1a shows 5 plants, but it is possible that some of the seedlings died after being transplanted into the soil); how many repetitions were used; how did bacteria inoculate plants?

A: Thanks for the comment. We have included more info about soil mix, general methodology, plants number, and so on. In respect to the loss of some seedlings, we clarify in the text that we wait at least 24 h since soil transplantation to ensure their viability. We didn’t specify, but if only one is lost, we replace it before inoculation to ensure the number per pot. As this could be worst, we were prepared with extra 5-plants pots to substitute those with many plants lost. We used to prepare half of the set as extra to ensure a minimal number or to use for propagation in case no loses. In general, we used not to lose more than 1-3 plants in the whole set, but this is something we keep doing to avoid problems.

- In the section Materials and methods, after paragraph 2.7 is followed by paragraph 2.9.

A: Thanks for the comment. Sorry for the mistake, now it is correct.

- Section 3.2. It is necessary to write a what do the abbreviations DEG and GO mean.

A: Thanks for the comment. Now it’s included.

- What does the sign "***" and the curved lines above the bars of the diagram in Fig. 3 mean?

A: Thanks for the comment. The meaning of the asterisks is now included in the legend. About the curved lines, not sure about your meaning, but maybe they are the comparison lines to help the understanding for each control.

- Line 293. "Some of these genes were even induced by more than 4-fold". What are these genes? In Fig. 3, there is only one such gene, Ð¥TH14.

A: Sorry, you’re right, my way to describe it was not totally correct. Now is changed in text. Thanks for the comment.

- Lines 312-313, 324-325, 338-339, 348-349. "This experiment was repeated three 312 times with similar results and n = 3 biologically independent samples." This phrase should be moved to the materials and methods section when describing the relevant methods.

A: Thanks for the comment. This was properly changed to M&M.

- Line 328. "The literature shows ...". Reference should be made to literature data.

A: Thanks for the comment. This phrase was quitted from this section because it is used in ‘Discussion’, where it has more sense, and it’s properly cited.

- Line 369-370. "...or an indirect action, such as the biocontrol of pathogens or the production of phytohormones". Why do the authors believe that the production of phytohormones by bacteria has an indirect rather than a direct effect on plants?

A: Thanks for the comment. Probably the original phrase was not the same, but this remained. Sorry for this mistake. Now it’s changed in the text.

- In the Discussion section, links to the authors' own drawings should be provided.

A: Thanks for the comment. We are not sure about the meaning of this comment, but we included more citations in some aspects that have not the back of one in the previous version

- In the list of references, there are many occasions when no journal titles in which articles were published are present (â„–â„– 3-5, 7, 10 16, etc.).

A: So sorry, probably a mistake during curating phase of the bibliography. In previous stages we didn’t detect this. Now all references are correct.

Round 2

Reviewer 2 Report

Accept in present form